# Prognostic Value of Measurable Residual Disease in Patients with AML Undergoing HSCT: A Multicenter Study

**DOI:** 10.3390/cancers15051609

**Published:** 2023-03-05

**Authors:** Teresa Caballero-Velázquez, Olga Pérez-López, Ana Yeguas Bermejo, Eduardo Rodríguez Arbolí, Enrique Colado Varela, Amparo Sempere Talens, María Belén Vidriales, María Solé-Rodríguez, Covadonga Quirós Caso, Estefanía Pérez López, Marta Reinoso Segura, Concepción Prats-Martín, Pau Montesinos, Jose A. Pérez-Simón

**Affiliations:** 1Department of Haematology, Instituto de Biomedicina de Sevilla (IBIS/CSIC), University Hospital Virgen del Rocío, Universidad de Sevilla, 41013 Seville, Spain; 2Department of Haematology, University Hospital Virgen del Macarena, 41009 Seville, Spain; 3Department of Haematology, Centro de Investigación del Cáncer (Instituto de Biología Molecular y Celular del Cáncer, CSIC-USAL), Instituto Biosanitario de Salamanca (IBSAL), Hospital Universitario de Salamanca, 37007 Salamanca, Spain; 4Laboratory Medicine Program, Department of Hematology, Hospital Universitario Central de Asturias, 33011 Asturias, Spain; 5Department of Haematology, CIBERONC, Instituto Carlos III, Hospital Universitario y Politécnico La Fe, 46026 Valencia, Spain; 6Department of Haematology, Hospital Juan Ramón Jiménez, 21005 Huelva, Spain; 7Laboratory Medicine Program, Department of Clinical Biochemistry, Hospital Universitario Central de Asturias, 33011 Asturias, Spain

**Keywords:** acute myeloid leukemia, AML, measurable residual disease, MRD, flow cytometry, stem cell transplantation

## Abstract

**Simple Summary:**

In patients diagnosed with acute myeloid leukemia (AML), relapse remains the main cause of mortality after allogeneic hematopoietic stem cell transplantation (HSCT). The detection of measurable residual disease (MRD) by multiparameter flow cytometry in AML patients undergoing HSCT is a powerful predictor of outcome. The aim of this study was performed a retrospective multicenter study to evaluate the prognostic value of MRD by second generation of MFC among patients undergoing HSCT, using recommendations from the Euroflow consortium. MRD levels prior to transplantation significantly influenced outcomes irrespective of the conditioning regimen. Positive MRD on day +100 after transplantation was associated with an extremely poor prognosis. Detection of positive MRD prior to and after transplantation performed with standardized technical conditions has prognostic value in real life.

**Abstract:**

Allogeneic hematopoietic stem cell transplantation (HSCT) represents the best therapeutic option for many patients with acute myeloid leukemia (AML). However, relapse remains the main cause of mortality after transplantation. The detection of measurable residual disease (MRD) by multiparameter flow cytometry (MFC) in AML, before and after HSCT, has been described as a powerful predictor of outcome. Nevertheless, multicenter and standardized studies are lacking. A retrospective analysis was performed, including 295 AML patients undergoing HSCT in 4 centers that worked according to recommendations from the Euroflow consortium. Among patients in complete remission (CR), MRD levels prior to transplantation significantly influenced outcomes, with overall (OS) and leukemia free survival (LFS) at 2 years of 76.7% and 67.6% for MRD-negative patients, 68.5% and 49.7% for MRD-low patients (MRD < 0.1), and 50.5% and 36.6% for MRD-high patients (MRD ≥ 0.1) (*p* < 0.001), respectively. MRD level did influence the outcome, irrespective of the conditioning regimen. In our patient cohort, positive MRD on day +100 after transplantation was associated with an extremely poor prognosis, with a cumulative incidence of relapse of 93.3%. In conclusion, our multicenter study confirms the prognostic value of MRD performed in accordance with standardized recommendations.

## 1. Introduction

Acute myeloid leukemia (AML) is a heterogeneous disease with different molecular and prognostic characteristics [1]. Allogeneic hematopoietic stem cell transplantation (HSCT) represents the best therapeutic option for many patients with intermediate- or high-risk AML [2]. However, relapse remains the main cause of mortality after transplantation.

The prognostic value of measurable residual disease (MRD) in AML patients is well recognized. Accordingly, 2017 European LeukemiaNet (ELN2017) introduced the new response category, complete remission (CR) without minimal residual disease [1]. The techniques used for MRD monitoring must be applicable, sensitive, specific, and reproducible. In this regard, quantitative polymerase chain reaction (PCR), and more recently, next generation sequencing and digital PCR, have a high sensitivity and applicability. However, they are limited to patients with certain genetic alterations, therefore, their applicability in routine clinical practice can be reduced. Multiparametric flow cytometry (MFC) is almost universally applicable, but display a lower sensitivity; the development of new generation flow (NGF) allows to achieve a similar sensitivity compared to molecular techniques [3]. Nevertheless, NGF is only validated in multiple myeloma and B acute lymphoblastic leukemia [4,5]. An elevated level of expertise is needed to perform MFC-MRD, and the harmonization of the technical issues for the measurement of MRD is necessary. According to ELN consensus, MRD by MFC must integrate the “leukemia-associated phenotype (LAP)” assessment, plus the “different from normal (DfN)” approach [1,6]. Other recommendations include considering the cut-off point of 0.1% to define MRD as positive, the use of prospectively validated panels, such as the Euroflow consortium [7], and the use of standardized flow cytometers [8]. However, there are no studies reporting routine clinical practice based on these standards.

In 2022, the update of the ELN classification emphasized the relevance of the early MRD evaluation that can modify the individual risk classification [9]. In addition, ELN 2017 and 2022 imply a broad genetic characterization, which is not considered in this study, since it includes patients transplanted from 2012 to 2020, who are therefore classified according to ELN 2011.

The detection of MRD by MFC in AML patients undergoing HSCT is a powerful predictor of outcome, and might allow individualized therapeutic strategies, as shown in different studies [10,11,12,13,14,15,16,17]. However, multicenter and standardized studies are lacking.

We performed a retrospective multicenter study to evaluate the prognostic value of MRD by second generation of MFC among patients undergoing HSCT, using recommendations from the Euroflow consortium.

## 2. Methods

### 2.1. Patients

A retrospective analysis was performed, including 295 AML patients treated according to PETHEMA protocols, undergoing HSCT from 2012 to 2020 in 4 transplant centers. CR was defined, based on ELN2011, as less than 5% of blasts in bone marrow (BM), and no evidence of extramedullary leukemia [17]. All patients who had flow cytometric evaluation prior to transplantation were included in the analysis. All patients provided written informed consent in accordance with the Declaration of Helsinki. This study has been approved by a formally constituted review board: C.P. S2200072, C.I. 0466-N-22, CEI de los Hospitales Universitarios Virgen Macarena y Virgen del Rocío (Protocolo V.1-21 February 2022).

### 2.2. Detection of Measurable Residual Disease by Second Generation Flow Cytometry

BM aspirate was obtained before starting the conditioning regimen, and at day one hundred after HSCT. The MRD was carried out using 8-color panels based on Euroflow protocols [7]. The Euroflow consortium periodically organizes an Internal Quality Assurance (QA) program to verify that the laboratories work in a standardized way, and according to the published recommendations. Two centers included in the study participate in the QA program organized since 2015, and another one from 2016. Appendix A shows the information from the QA program. All the centers carry out calibration and stability controls of the cytometers daily, with CS&T (Becton Dickinson, San Jose, CA, USA) and SPHEROTM Rainbow Calibration Particles, EuroFlowTM. If it fails, and failure is not due to a fluid problem, the technical service is notified. Normal cells were used for internal control, case by case. Fifty µL of BM per tube were stained with monoclonal antibodies. After 30 min of incubation at room temperature in the dark, erythrocytes were lysed, and the sample was washed. The samples were acquired in 8-color digital cytometers FACSCanto II (Becton Dickinson, San Jose, CA, USA). Cytometers were calibrated and compensated according to Euroflow protocols using the Diva software (Becton Dickinson) [8]. More than five hundred thousand viable cells per tube were acquired. For the design of the MRD, the different laboratories used panels from the myelodysplastic syndrome (MDS) panel of Euroflow (T1, T2, and T3 in more than 80% ± T4 when lymphoid markers were expressed, or when the phenotype at diagnosis was not available), along with additional tubes, depending on the patient’s specific “leukemia-associated phenotype” (LAP). The tubes that include the LAP had to include the 4 backbones, HLADR, CD45, CD34, and CD117, together with another 4 antibodies at the discretion of each laboratory, in accordance with the recommendations by ELN for MRD [6]. In 240 cases, tubes were added to the maturation tubes of the Euroflow MDS diagnostic panel. The combinations for LAP analysis are shown in Appendix A. When the phenotype of the blasts at diagnosis corresponded to monocytes, more than 1 tube associated with LAP was designated.

Analysis was performed with Infinicyt software 1.6 to 2.0 (Cytognos). Both LAP assessment plus the “different from normal (DfN)” approach were used to analyze MRD according to the criteria of each laboratory. Any measurable MRD level was considered as positive. A representative MRD image is shown in Appendix A. The abnormal population was quantified as a percentage of the total viable cells (including the erythroblast population). There was no reference laboratory, but each laboratory independently carried out the analysis of its samples. All centers carried out the same initial strategy of population selection analysis based on the 4 common markers. The 4 centers confirmed the MRD in more than 1 tube. MRD was considered assessable when the peripheral blood cell counts were recovered. Maturation patterns were analyzed. MRD assays were performed by 2 experts, and most included the lead analyst. A comparative study of 6 MRD samples was carried out between the 4 centers (Appendix A). MRD included in this study achieved a sensitivity of 0.1%, and more than 90% was 0.01%. The level of sensitivity was considered based on the following conditions: number of acquired events, viable cells, patient’s LAP if available, and bone marrow status (representation of bone marrow cells: mast cells, plasma cells, B precursors...). We considered the marrow evaluable for analysis if mast cells, red series, and less than 80% mature neutrophils were found. A cluster of 20 cells with phenotypic abnormalities was needed for the detection of MRD.

### 2.3. Statistical Analysis

The objective of our study was to evaluate the impact of MRD on the outcome of patients with AML undergoing HSCT. The following endpoints were analyzed: cumulative incidence of relapse (CIR), acute and chronic graft versus host disease (GvHD), non-relapse mortality (NRM), overall survival (OS), and leukemia free survival (LFS). Probabilities of OS and LFS were calculated using the Kaplan-Meier method. Cumulative incidence functions were used to estimate CIR, NRM, aGvHD, and cGvHD rates in the setting of competing risks. OS was calculated from the time of transplantation to death from any cause, and those who survived were censored at last follow-up. LFS was calculated from the time of transplantation to relapse or death, and those patients who did not obtain a CR were considered events. Neutrophil recovery was considered as more than 0.5 × 10^9^/L for at least 2 consecutive days, and platelet engraftment was more than 20 × 10^9^/L platelets for at least 2 consecutive days. Comparisons were performed using the log-rank test for LFS and OS, and Gray’s test for RI and NRM. The impact of age, type of conditioning, GvHD prophylaxis, ELN classification, and MRD (according to ELN criteria) was evaluated both in univariate and multivariate analysis. Cox proportional hazards regression models were constructed for OS and LFS. GvHD was diagnosed according to NIH criteria. The occurrence of chronic GvHD was treated as a time-dependent covariate. Patients who lived more than 100 days after transplantation were evaluated for chronic GvHD. Cumulative incidences were calculated with the cmprsk package for R version 2.14.0, and other analyses were performed using SPSS 20.0 and Stata 14.2.

## 3. Results

### 3.1. Patient Characteristics

A total of 295 patients were included into the analysis; Table 1 shows the characteristics of the patients. Of them, 285 (96.7%) were in CR at the time of HSCT, 207 had negative MRD (MRD neg), in 21 patients MRD was positive but below 0.1% (namely MRD-low), and in 57 patients MRD was positive and greater than or equal to 0.1% (MRD-high or MRD ≥ 0.1). Ten patients had active disease. A total of 47.1% received HSCT from a matched sibling donor, while 39.7% had unrelated donors. A total of 59.7% of patients received myeloablative conditioning. Considering only patients who were in CR at the time of transplant, no significant differences were observed between patients with MRD positive (MRD pos) or negative, except for age, the prior being slightly older (54.5 vs. 51 years, *p* = 0.02), and GvHD prophylaxis, with MRD pos patients receiving more T-cell depletion-based approaches (Table 1).

### 3.2. Transplant Toxicities and GvHD

Regarding the toxicity of the procedure, no patient in the MRD pos, and three in the MRD neg group, had neutrophil graft failure. Regarding neutrophil engraftment, no differences were observed between MRD neg and MRD pos patients: median of 16 days (range 8–385) among MRD pos patients, versus 16 days (range 9–181) for MRD neg patients. Regarding platelet engraftment, two patients in both MRD pos and MRD neg groups did not engraft. There were no differences in the speed of platelet engraftment between MRD pos (15, range 5–171) versus MRD neg patients (12, range 3–1096) (Table 2).

One hundred and eighty-four (62.4%) patients developed acute GvHD: 56 presented grade one, 100 grade two, 15 grade three, and 12 grade four. Chronic GvHD was observed in 121 patients (41%); in 60 (20%) of them it was mild, and in 61 (20.7%) it was moderate/severe. There were no differences in the incidence of acute or chronic GvHD between MRD pos and MRD neg groups (Table 2).

On the other hand, 76 MRD positive patients with were evaluable for acute GvHD, and 32 of them relapsed. Of the patients, 63.5% (28 of 44) who did not relapse developed acute GvHD, versus 40.6% (13 of 32) of patients who relapsed (*p* = 0.063). In other words, we observed a trend towards a higher risk of aGvHD in MRD positive patients who did not relapse, as compared to those who did. No significant differences in chronic GvHD were observed: 61.3% (27 of 44) of patients who did not relapse developed chronic GvHD, versus 71.9% (23 of 32) of patients who did (*p* = 0.5).

### 3.3. Impact of MRD Prior to Transplantation on Outcomes after HSCT

Overall survival at 2 and 5 years was 69% [95% CI 63.18–74.18] and 58.4% [95% CI 52.4–63.9], respectively, and the LFS at 2 and 5 years was 57.8% [95% CI 50.8−64.7%] and 50.4% [95% CI 44–57], respectively. Among patients in CR, MRD levels significantly influenced outcomes, with OS and LFS at 2 years of 76.7% [95% CI 70.1–82] and 67.6% [95% CI 60.6–73.6] for MRD neg patients, 68.5% [95% CI 42.1–84.7] and 49.7% [95% CI 26.4–69.3] for MRD-low patients, and 50.5% [95% CI 36–63.2] and 36.6% [95% CI 23.7–49.5] for MRD-high patients, respectively (MRD ≥ 0.1) (*p* < 0.001) (Appendix A). Next, we attempted to confirm the prognostic value of the ELN17 proposed cut-off (0.1%) to identify patients at a higher risk of relapse and, therefore, we combined the MRD neg and MRD-low groups into MRD < 0.1: at 2 and 5 years, the respective values for these MRD < 0.1 patients were 76% [95% CI 69.7–81.1] and 66.7% [95% CI 59.1–73.1] for OS (*p* < 0.001 as compared to those with MRD ≥ 0.1), and 66% [95% CI 59.3–71.8] and 58.1% [95% CI 50.6–64.8] for LFS (*p* < 0.001 as compared to MRD ≥ 0.1), respectively. In Figure 1, the OS and LFS of the three groups MRD < 1, MRD ≥ 0.1, and active disease are shown (*p* < 0.001).

In univariate analysis, age (HR = 1.03 [95% CI 1.01–1.04], *p* = 0.002 and HR = 1.02 [95% CI 1.01–1.03], *p* = 0.008), conditioning (HR = 1.73 [95% CI 1.18–2.53], *p* = 0.005 and HR = 1.43 [95% CI 1.01–2.01], *p* = 0.041), adverse risk of ELN2011 (HR = 3.26 [95% CI 1.62–6.55], *p* = 0.001 and HR = 3.75 [95% CI 1.98–7.09], *p* < 0.001), and MRD ≥ 0.1 (HR = 2.51 [95% CI 1.64–3.83], *p* < 0.001 and HR = 2.29 [95% CI 1.55–3.38], *p* < 0.001) significantly influenced OS and LFS, respectively (Appendix A).

In multivariate analysis, adverse risk of ELN2011 (HR = 2.42 [95% CI 1.18–5.00], *p* = 0.016 for OS and HR = 3.16 [95% CI 1.64–6.07], *p* = 0.001 for LFS) and MRD ≥ 0.1 (HR = 2.07 [95% CI 1.26–3.39], *p* = 0.004 for OS and HR 2.1 [95% CI 1.36–3.29], *p* = 0.001 for LFS) significantly influenced OS and LFS (Table 3).

In multivariate time-dependent analysis (time-dependent variable GvHD), the adverse risk group, according to ELN2011 (for OS HR= 2.8 95% [CI 1.33–5.89], *p* = 0.006 and for LFS HR = 3.43 [95% CI 1.76–6.7], *p* < 0.001), and MRD ≥ 0.1 before transplant (for OS HR = 1.98 [95% CI 1.26–3.11], *p* < 0.001 and for LFS HR = 1.89 [95% CI 1.25–2.86], *p* < 0.001) significantly influenced the outcome (Appendix A).

Next, we attempted to identify whether these differences, in terms of OS and LFS between MRD ≥ 0.1 and MRD < 0.1, were due to a higher RI and/or NRM.

Considering only patients in CR, CIR at 2 and 5 years were significantly lower among MRD < 0.1 patients: 22% [95% CI 17–28.1%] and 27% [95% CI 21−33.5%], versus 46.5% [95% CI 32.4−59.5%] and 50% [95% CI 34.8−63.2%] among MRD ≥ 0.1 patients, respectively, *p* = 0.0005.

By contrast, no differences were observed in terms of NRM (11.7% [95% CI 7.9−16.3%] and 14.8% [95% CI 10.1–20.4%] for MRD < 0.1, and 16.9% [95% CI 8.2−28.3%] and 23.7% [95% CI 11.9−37.8%] for MRD ≥ 0.1 *p* = 0.2), (Figure 2).

In univariate analysis, age (HR = 1.02 [95% CI 1–1.04], *p* = 0.037), adverse risk of ELN2011 (HR = 4.88 [95% CI 1.99–11.99], *p* = 0.01), and MRD ≥ 0.1 (HR = 2.29 [95% CI 1.43–3.65], *p* = 0.001) significantly influenced CIR. As far as NRM is concerned, only the type of donor had a statistically significant influence in univariate analysis: unrelated donor (HR = 2.04 [95% CI 1.06–3.95], *p* = 0.033) and haplo-identical donor (HR = 2.73 [95% CI 1.16–6.42], *p* = 0.021) (Appendix A).

Finally, in multivariate analysis, haplo-identical donor (HR = −0.27 [95% CI 0.11–0.67], *p* = 0.005), T-cell depletion (HR 1.78 [95% CI 1.01–3.14], *p* = 0.045), adverse risk ELN2011 (HR = 4.37 [95% CI 1.67–11.4], *p* = 0.003), and MRD ≥ 0.1 (HR = 2.47 [95% CI 1.4–4.33], *p* = 0.002) significantly influenced CIR. On the other hand, the use of haplo-identical donor (HR = −2.63 [95% CI 1.04–6.65], *p* = 0.042) significantly influenced NRM (Table 3).

In Appendix A, we describe 43 patients who had MRD pos and did not relapse. Of these 43 patients, 30 (70%) developed acute GvHD (70%): 15 grade two, 4 grade four, and the rest grade one. Nineteen developed chronic GvHD (44.2%). Fourteen patients had MRD < 0.1, and three of them died because of an infection within one year after transplantation. The remaining 30 had MRD ≥ 0.1 and 11 (37%) of them have died, 8 before one year post-transplant because of infections (2 bilateral pneumonia), 2 due to GVHD, 1 due to veno-occlusive disease, 1 due to ureteral carcinoma, and another of unknown cause. Four of these thirty patients are still alive with a follow-up of less than 1 year.

### 3.4. Impact of MRD before Transplantation on OS and LFS among the Different ELN2011 Subgroups and Conditioning Regimens

MRD before transplantation also identified patients with different outcomes within the ELN2011 subgroups: OS and LFS at 2 years among high-risk ELN2011 patients were 58% and 41.1% in MRD < 0.1 vs. 39.3% and 20% for MRD ≥ 0.1 patients, *p* = 0.16 and *p* = 0.216, respectively; for intermediate risk: 77.6% and 70.2% for MRD < 0.1 vs. 60.6% and 43.5% among MRD ≥ 0.1 patients, *p* = 0.0.018 and *p* = 0.0037, respectively; and for favorable ELN2011 risk: 88.6% and 82.9% among MRD < 0.1 vs. 48.5% and 49.1% for patients with MRD ≥ 0.1, *p*= 0.0098 and *p*= 0.0315, respectively (Figure 3).

Finally, MRD level did influence the outcome, irrespective of the conditioning regimen: patients with MRD < 0.1 before transplantation had a better OS and LFS at 2 years (82% and 71.4% among those who received myeloablative conditioning, and 65% and 57.6% among those who received reduced intensity conditioning, respectively) than those who had MRD ≥ 0.1 prior to transplantation (56% and 44.4% for myeloablative, and 43% and 25.5% for those who received reduced intensity conditioning) (*p* < 0.001) (Figure 4).

### 3.5. Impact of MRD on 100 Days after HSCT

Next, we analyzed the prognostic value of MRD at 100 days post transplantation; at 2 years, OS and LFS were significantly higher in patients with MRD < 0.1 on day +100 (83% and 76% in MRD < 0.1 vs. 76% and 7% in MRD ≥ 0.1, respectively, *p* < 0.001) (Figure 5). Similarly, CIR at 2 years was significantly lower in patients with MRD < 0.1 (12.8% [95% CI 8.4–18%] as compared to 93.3% [95% CI 41.4%−99.5%]) among those with MRD ≥ 0.1, *p* < 0.001. By contrast, no differences were observed in terms of NRM (10.6% [95% CI 6.7–15.6] in MRD < 0.1 vs. 0% in in MRD ≥ 0.1, *p* = 0.19).

However, the main limitation for the analysis of the MRD at 100 days after transplantation is that only 16 patients with positive MRD remained in CR at this time-point.

## 4. Discussion

Numerous studies have described the prognostic value of measurable residual disease after induction and/or consolidation [18,19,20,21,22,23,24,25,26,27] and, based on these results, it is currently used to tailor the intensification strategy in patients with AML [21]. Terwijn et al. showed the independent prognostic value of MRD ≥ 0.1% by MCF after induction and consolidation therapy, identifying a subgroup of patients with higher risk of relapse [22]. Similarly, a second multicenter prospective study identified that patients with positive MRD had a poor outcome [23]. On the other hand, Balsat et al. showed the prognostic value of the detection of mutated NPM1 in peripheral blood after induction, thus identifying a subgroup of patients who would be candidates for transplantation [24]. Likewise, *RUNX1-RUNX1T1* MRD monitoring also identifies a subgroup of patients at high risk of relapse [25]. Moreover, Jongen-Lavrencic et al. showed that the combination of detection of MRD by next-generation sequencing and flow cytometry has additional prognostic value in terms of relapse and overall survival [26]. Based on these findings, Cornelissen et al. proposed an algorithm, considering not only the risk subgroup at diagnosis and the performance status of the patients, but also the levels of MRD [27].

Unfortunately, although HSCT is considered the best option for patients with persistent MRD after first line treatment, tumor load, even at the MRD level, is one of the variables with the highest impact on outcome, after allogeneic stem cell transplantation, as described in different studies using either molecular or flow cytometry techniques. Regarding the latter, available studies have not used international standardized protocols; different cut-off values have been used, and the number of combinations of the fluorochrome-conjugated antibodies used, or the analysis strategy, have also greatly varied. Likewise, the need to maintain calibration and compensation protocols is not considered. Araki et al. showed that patients in CR with positive MRD at the time of transplantation had a similar outcome compared to patients with active disease with OS at 3 years, of 26% and 23%, respectively [14]. In this study, 1 million events were acquired, and a panel of 10 colors with three combinations of antibodies is carried out, considering any level of detection to determine positive MRD. However, it is a single center study conducted by the same group of hematopathologists. Additionally, a different from normal approach is used, considering a population showing a deviation from antigen expression patterns on reactive or regeneration hematopoietic cells. However, this approach might not display a uniform sensitivity in all cases. Moreover, it is not considered an international standardization of cytometers, allowing an adequate reproducibility within different laboratories. Likewise, in the same group in the context of myeloablative transplantation, the presence of MRD before or after transplantation identified patients with poor outcome [13] As in our study, they observed that 16 patients with MRD pos after transplantation had shorter OS and RFS. However, the MRD measurement is performed at an earlier post-transplant period, between days +28 and +35. Other studies that show the prognostic value of positive MRD use numbers of fluorescence less than six [17,28,29], or combine the detection of disease with low-sensitivity techniques, such as karyotype or fluorescence in situ hybridization (FISH) [28]. In a recent retrospective study, patients with monosomal karyotype had a higher likelihood of MRD positivity by MFC prior to transplant, with worse OS and higher relapse risk than those without it [12]. However, in multivariate analysis, only MRD positivity was associated with a shorter survival.

The current study has been conducted in routine clinical practice, in real life, outside of a specific trial, in four centers that had to meet certain requirements. Three laboratories participated in the quality control organized within the Euroflow QA program. Two different approaches were used to determinate MRD: “leukemia-associated phenotype (LAP)” and “different from normal (DfN)”. The latter allowed us to identify MRD even if the diagnostic phenotype was not available, or in case of changes in the phenotype during the disease [30,31,32]. Both strategies have been used, as recommended by ELN [1,6]. All laboratories work with the same quality criteria, and each one carried out its own analyses by experts in clinical cytometry.

In the current study, we confirmed the prognostic value of MRD monitoring, both pre- and post-allogeneic stem cell transplantation. In fact, cumulative incidence of relapse was significantly higher, both if we consider patients with MRD pos as compared to MRD neg, and by using the cut-off value suggested by ELN, and this difference in relapse incidence significantly influenced OS and LFS.

Considering ELN2011 subgroups, MRD also allowed the identification of subgroups of patients with different prognoses among favorable and intermediate ELN2011, while in the adverse risk group, no clear difference was observed. The number of patients included in each subgroup, when the analysis is separately performed, might explain the lack of statistically significant differences in this subgroup, in terms of overall survival. Alternatively, the poor outcome of these patients might not be influenced by the MRD levels. Further studies would be required to elucidate this point.

An area of current interest, in the context of allogeneic transplantation, is whether the therapeutic strategy should be modified in those patients with positive MRD, before transplantation. In fact, several studies have suggested that patients with positive MRD prior to transplant might benefit from receiving myeloablative conditionings [17,33,34]. However, in a recent retrospective study of 746 patients, Morsink et al. observed that in all patients, regardless of the type of conditioning, the risk of relapse, relapse free survival, and OS were higher in patients with positive, as compared to those with negative, MRD [11]. Similarly, no effect of the conditioning regimen intensity on OS was observed in *NPM1* mutated AML patients who remained MRD positive [35]. In the same way, in the current analysis, regardless of the type of conditioning, the presence of MRD by flow cytometry implied a poor prognosis. Recently, Paras et al. conducted a retrospective study of 810 patients, showing that the conditioning regimen can influence the ability to eliminate MRD; however, the elimination of MRD in non-myeloablative regimens had a greater impact on the outcome [36]. With these contradictory results, it remains unclear whether MRD should determine the choice of conditioning regimen. An alternative approach would be to identify these patients for alternative therapy after HSCT, such as withdraw of immunosuppression, hypomethylating agents [37], infusion of donor lymphocytes [38], or venetoclax [39], among other approaches. For example, the RELAZA2 prospective study demonstrated that treatment with azacytidine [37] in patients with MRD pos could avoid or delay hematology relapse of AML or high-risk myelodysplastic syndromes (MDS). In this regard, MRD monitoring after transplantation would further allow the identification of patients at a high risk of relapse, and establish therapeutic maneuvers.

In our patient’s cohort, positive MRD on day +100 after transplantation was associates with an extremely poor prognosis, with a CIR of 93.3%. It is possible that the identification of MRD should be conducted in an earlier post-transplant period, to assess any preventive strategy to avoid relapse. Moreover, in our study, those patients who reach day +100 post-transplant and persist, or have a new positive MRD, have an unfavorable short-term prognosis, regardless of the status of the pre-transplant MRD. In this way, a recent study of Paras et al. showed that the dynamics of MRD before, and early after transplantation (+20 or 40 days after transplantation), improve the accuracy of risk assessment; in that study, patients with MRDpos/MRDpos and MRDneg/MRDpos had high relapse rates and poor survival estimates [36].

In relation to the type of donor, is relevant to point out that Yu et al. have shown that a haplo-identical donor might increase the graft versus leukemia effect [40]. Therefore, the positivity of MRD before transplant could modify the selection of the donor.

Other authors have identified the CD34+/CD38– leukemic stem cell (LSC) phenotype, and have shown its prognostic value, both at diagnosis and during follow-up [41,42,43,44]. Furthermore, this phenotype might be used in combination with the MRD assessment in a single tube combining six markers [43]. Therefore, future studies might allow the further improvement of MRD analysis, for example, by incorporating the LSC phenotype into the standardized MRD strategy currently used.

The main limitation of this study is its retrospective nature over a long period of time, in which changes have been incorporated to the treatment according to the molecular results. Likewise, GvHD conditioning and prophylaxis vary during this period. On the other hand, each center carried out its analysis independently. However, this further reinforces the fact that a positive MRD performed with standardized technical conditions has prognostic value in real life.

## 5. Conclusions

Our multicenter study confirms the prognostic value of MRD performed in accordance with standardized recommendations. Regardless of the conditioning regimen, positive MRD maintains its poor prognostic value, and might allow the identification of patients who are candidates to receive early post-transplant therapeutic procedures.

## Figures and Tables

**Figure 1 cancers-15-01609-f001:**
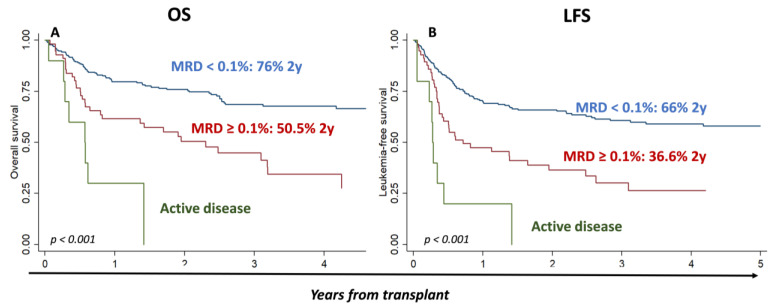
Impact of MRD levels in OS and LFS, considering MRD < 0.1%, MRD ≥ 0.1%, and group of active disease.

**Figure 2 cancers-15-01609-f002:**
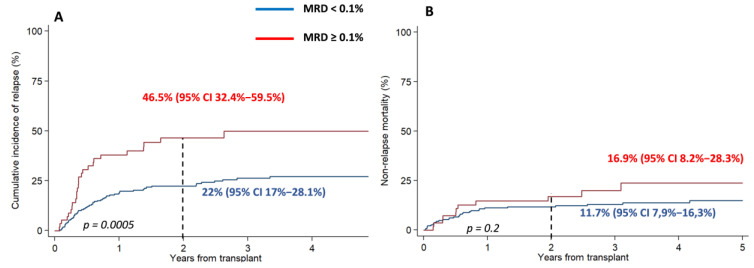
Impact of MRD in CIR and NRM: (**A**) CIR at 2 and 5 years were significantly lower among those with MRD ≥ 0.1. (**B**) However, no differences were observed in terms of non-relapse mortality.

**Figure 3 cancers-15-01609-f003:**
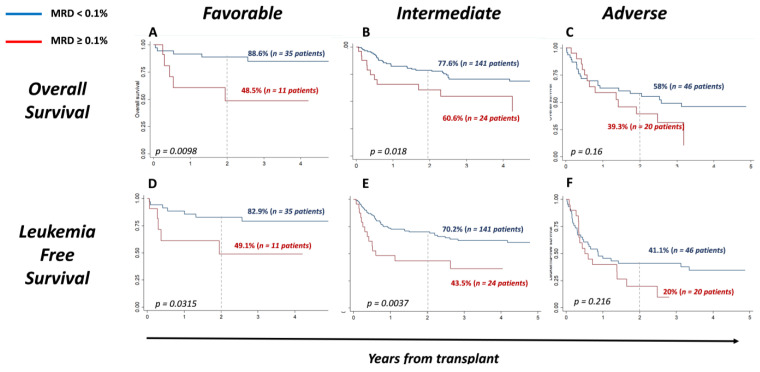
Impact of MRD in OS and LFS according to risk group ELN2011: survival was significantly worse in patients with MRD ≥ 0.1 in favorable (**A**,**D**) and intermediate groups (**B**,**E**). However, no differences were observed in the adverse risk group (**C**,**F**).

**Figure 4 cancers-15-01609-f004:**
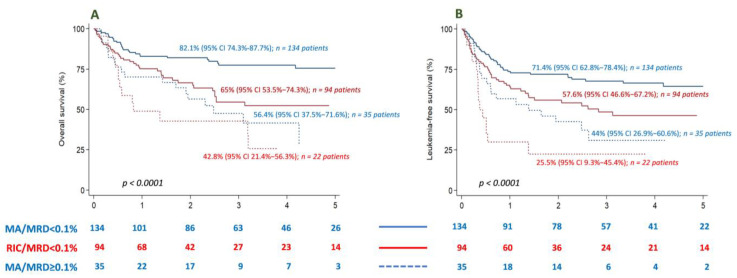
Impact of MRD in OS and LFS, according to conditioning regimen. Patients with MRD < 0.1 before transplantation had a better OS (**A**) and LFS (**B**) at 2 years, with both types of conditioning.

**Figure 5 cancers-15-01609-f005:**
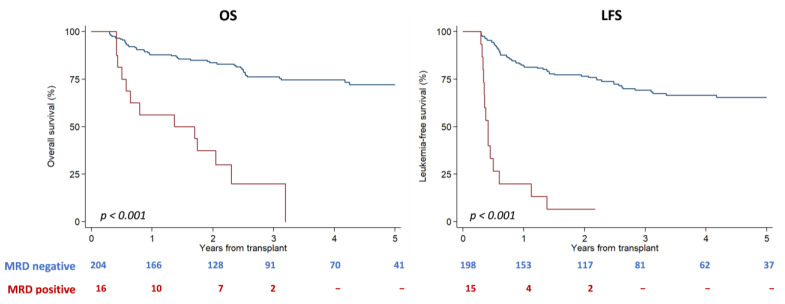
Impact of MRD on day +100 after transplantation. At 2 years, OS and LFS were significantly higher in patients with MRD ≥ 0.1 on day +100.

**Table 1 cancers-15-01609-t001:** Characteristics of the patients according to MRD before transplantation.

	All Group (*n* = 295)	MRD Negative (*n* = 207)	MRD Positive (*n* = 78)	*p*-Value
AGE mean (range)	51 (2–71)	51 (2–71)	54.5 (18–69)	0.02
Recipient Gender, *n* (%) Male	148(50.2)	102(49.3)	40(51.3)	0.63
ELN2011, *n* (%) favorable intermediate I intermediate II adverse	47 (15.9) 57 (19.3) 111 (37.6) 70 (23.7)	31 (15) 48 (23.2) 79 (38.2) 44 (21.3)	15 (19.2) 8 (10.3) 30 (38.5) 21 (28.2)	0.217
Disease Status at transplant, *n* (%) 1st CR 2nd CR Others CR Active disease Aplasia	235 (79.7) 28 (9.5) 19 (6.4) 10 (3.4) 3 (1)	176 (85) 17 (8.2) 13 (6.3) 1 (0.5) 0	59 (77.6) 11 (14.5) 6 (7.9) 0 2	0.271
Donor, *n* (%) Matched sibling Unrelated donor Haplo-identical donor	139 (47.1) 117 (39.7) 38 (12.9)	103 (49.8) 82 (39.6) 22 (10.6)	31 (39.7) 30 (38.5) 16 (20.5)	0.092
Conditioning, *n* (%) Myeloablative No myeloablative	176 (59.7) 117 (40.4)	119 (57.5) 88 (42.5)	50 (64.1) 28 (35.9)	0.781
Conditioning Therapy, *n* (%) BUCy BUCy + Thiothepa FLUBU FLUBU + Cy FLUBU + THIOTHEPA FLUBU + THIOTHEPA + ATG FLUBU + ATG Cy TBI ± ATG FLUMEL ± Thiothepa Others	56 (19) 3 (1) 162 (54.9%) 8 (2.7%) 42 (14.2) 2 (0.7) 2 (0.7) 5 (1.3) 7 (2.3) 8 (2.7)	44 (21.3) 1 (0.5) 113 (54.6) 5 (2.4) 33 (15.9) 2 (1) 1 (0.5) 2 (1) 2 (1) 4 (1.9)	10 (12.8) 0 (0) 44 (56.4) 3 (3.8) 9 (11.5) 0 (0) 1 (1.3) 2 (2.6) 5 (6.4) 4 (5.2)	0.385
Donor gender, *n* (%) Male	192 (65.1)	135 (65.2)	51 (65.4)	0.945
Donor/Recipient gender, *n* (%) Female/male	52 (17.6)	38 (18.4)	12 (15.4)	0.807
GvHD Prophylaxis, *n* (%) Tacrolimus/CsA + MTX Tacrolimus + Sirolimus ± MMF Tacrolimus/CsA + MTX + ATG Tacrolimus/CsA + MMF Tacrolimus/CsA + MMF + Cy Sirolimus + MMF + Cy CsA + Pred	121 (41) 82 (27.8) 28 (9.5) 25 (8.5) 24 (8.1) 10 (3.4) 2 (0.7)	87 (42) 64 (30.9) 14 (6.7) 18 (8.7) 13 (6.2) 8 (3.9) 2 (1)	28 (35.9) 16 (21) 13 (16.6) 7 (9) 11 (14.1) 2 (2.6) 0 (0)	0.001

ELN: European LeukemiaNet; CR: complete remission; BUCy: busulfan + cyclophosphamide; FLUBU: fludarabine + busulfan; Cy: cyclophosphamide; ATG: thymoglobulin; TBI: total body irradiation; MTX: methotrexate; MMF: mycophenolate mofetil; CsA: cyclosporine A; GvHD: graft versus host disease.

**Table 2 cancers-15-01609-t002:** Toxicities: engraftment and GvHD.

	All Group (*n* = 295)	MRD Negative before Transplantation (*n* = 207)	MRD Positive before Transplantation (*n* = 78)	*p*-Value
Engraftment (YES/patients) Neutrophil Platelets	286/289 287/293	204/207 205/207	78/7 876/78	
Engraftment day mean (range) Neutrophil Platelets	16 (8–385) 13 (3–1096)	16 (9–181) 12 (3–1096)	16 (8–385) 15 (5–171)	0.889 0.317
Acute GvHD, *n*(%) Grade 1 Grade 2 Grade 3 Grade 4	184 (62.4) 56 (19) 100 (33.9) 15 (5.1) 12 (4.1)	137 (55.6) 45 (21.7) 71 (34.3) 12 (5.8) 8 (3.9)	43 (55.1) 11 (14.1) 28 (36.8) 0 (0) 3 (3.9)	0.137
Chronic GvHD, *n* (%) Mild Moderate/severe	121 (59) 60 (20) 61 (20.7)	92 (44.4) 47 (22.7) 41 (19.8)	28 (35.9) 13 (16.7) 14 (17.9)	0.356

MRD: measurable residual disease; GvHD: graft versus host disease.

**Table 3 cancers-15-01609-t003:** Multivariate analysis considering MRD before transplantation.

Variable	CIR HR (95% CI)	NRM HR (95% CI)	OS HR (95% CI)	LFS HR (95% CI)
Sex male	0.93 (0.59–1.48), *p* = 0.785	1.0 (0.53–1.92), *p* = 0.980	1.05 (0.68–1.60), *p* = 0.837	1.01 (0.7–1.47), *p* = 0.852
Age	1.01 (0.98–1.04), *p* = 0.456	1.02 (0.99–1.05), *p* = 0.231	1.02 (1.00–1.04), *p* = 0.077	1.02 (0.99–1.04), *p* = 0.065
Conditioning RIC	1.17 (0.65–2.10), *p* = 0.595	1.32 (0.62–2.84), *p* = 0.467	1.65 (0.95–2.88), *p* = 0.077	1.32 (0.82–2.12), *p* = 0.248
Donor Unrelated Haploidentical	1.04 (0.64–1.69), *p* = 0.886 0.27 (0.11–0.67), *p* = 0.005	2.05 (0.94–4.50), *p* = 0.073 2.63 (1.04–6.65), *p* = 0.042	1.25 (0.79–1.99), *p* = 0.333 0.86 (0.43–1.69), *p* = 0.655	1.29 (0.87–1.94), *p* = 0.206 0.69 (0.37–1.29), *p* = 0.248
Depletion T Yes	1.78 (1.01–3.14), *p* = 0.045	0.56 (0.26–1.21), *p* = 0.138	0.94 (0.56–1.57), *p* = 0.812	1.13(0.73–1.76), *p* = 0.583
ELN_2011 Intermediate Adverse	2.51 (0.99–6.34), *p* = 0.052 4.37 (1.67–11.4), *p* = 0.003	1.09 (0.43–2.82), *p* = 0.851 1.45 (0.53–4.00), *p* = 0.470	1.51 (0.75–3.04), *p* = 0.243 2.42 (1.18–5.00), *p* = 0.016	1.88 (1.00–3.54), *p* = 0.05 3.16 (1.64–6.07), *p* = 0.001
MRD ≥ 0.1 before transplantation	2.47 (1.4–4.33), *p* = 0.002	1.10 (0.47–2.59), *p* = 0.821	2.07 (1.26–3.39), *p* = 0.004	2.1 (1.36–3.29), *p* = 0.001

CIR: cumulative incidence of relapse; NRM: non-relapse mortality; OS: overall survival; LFS: leukemia free survival; HR: hazard ratio; CI: confidence interval; Mielo: myeloablative; RIC: reduced intensity conditioning; ELN: European LeukemiaNet; MRD: measurable residual disease.

## Data Availability

The original data will be available by request to the corresponding author.

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
