# Peer review of "Prognostic Value of Measurable Residual Disease in Patients with AML Undergoing HSCT: A Multicenter Study"

_cancers, 2023, doi:10.3390/cancers15051609_

Round 1
Reviewer 1 Report (New Reviewer)
The authors here present a real-life analysis regarding the impact of MRD assessment prior and 100 days after hematopoietic stem cell transplantation in AML patients. Although the data refer to patients treated between 2012 and 2020, which implies some limitations, the message is of interest for the hematological community.
The reviewer recommends to mention in the introduction section the importance reserved to MRD in the new ELN 2022 guidelines and also mention, among limitations, the lack of an extensive molecular analysis that could have improved patients' classification according to ELN 2017 or ELN 2022 indications.
Author Response
In accordance with the reviewer's recommendation, the following comment has been included in the instruction section:
In 2022, the update of the ELN classification emphasizes the relevance of the early MRD evaluation that can modify the individual risk classification(9). In addition, ELN 2017 and 2022 imply a broad genetic characterization, which is not considered in this study since it includes patients transplanted from 2012 to 2020, therefore classified according to ELN 2011.
Reviewer 2 Report (New Reviewer)
Teresa Caballero-Velázquez et al. analyzed the relationship of measurable residual disease (MRD) detection and outcomes of AML patients underwent allogeneic hematopoietic stem cell transplantation (HSCT). The manuscript was organized well and the data are convincing. Unfortunately, most the conclusions in this manuscript were published elsewhere and the authors didn’t provide many interesting and novel findings. Besides, the reviewer has some minor issues listed below:
1. Why did the authors lyse erythroid after incubation with antibodies (Line 90-91)? Lysing erythroid before incubation will reduce the potential disturbance of erythroid cells.
2. Line 133, “9” should be sup script.
Author Response
- Why did the authors lyse erythroid after incubation with antibodies (Line 90-91)? Lysing erythroid before incubation will reduce the potential disturbance of erythroid cells.
According to the reviewer, the MRD Euroflow protocols of B-acute lymphoblastic leukemia and multiple myeloma performed bulk lysis, that is lyse prior to incubation with antibodies. However, between the recommendation of MRD form ELN2021 you could prepare the sample by two ways: bulk lysis or stain-lyse-wash or no wash. This study started in 2012 when the articles of Euroflow were published and we performed the standard operating procedures for diagnosis developed by Euroflow consortium.
- Line 133, “9” should be sup script.
Thank you for your commentary, we have corrected those mistakes.
Reviewer 3 Report (Previous Reviewer 1)
The authors have provided more information about the applied methods, which helps to understand how the research was performed. In the paper, the authors explicitly mention the need for multicenter and standardized studies several times (eg in Abstract, Introduction and Discussion), but if they feel this is so important they also must provide the corresponding data, or they should remove these statements.
Therefore my main concerns/remarks are still:
- The EuroFlow AML/MDS panel is not designed for MRD. If the authors use it for MRD analysis, they must show data on the sensitivity. How was the sensitivity defined? How often was 0,1% reached, or 0,01%? Also it should be indicated how often tube 1 to 4 was used or if in some patients only a limited number of tubes was used (and if so, why).
- Next to the EuroFlow tubes they use a LAIP tube. The authors should indicate which other markers (next to the backbone) are generally used (e.g. in a Supplemental Table).
- The authors use a DfN and LAIP approach. It would be relevant and informative to show how these two approaches compare to each other. If the two approaches give discordant results, which data was used (e.g. highest of the two)?
- If the authors focus on the multicenter aspect, they must provide data showing that the data from the 4 centers are comparable. They could for example show that patients characteristics are similar between the four centers and that also the distribution of MRD levels was similar between the four centers. Obviously it would be best if they could show that similar results were obtained in QA rounds, either based on fresh samples or electronic data files. Since centers used standardized protocols re-analysis of electronic data files should be relatively easy.
- Related to the above, it is still not clear how reproducible the data analysis is, adding such data would be very helpful.
- Since the EuroFlow AML/MDS panel is not designed for MRD, the authors should not refer to it as "according to recommendations from the EuroFlow consortium". They may indicate that instrument settings were performed according to EuroFlow and the the EuroFlow AML/MDS panel was used. It should also be noted that EuroFlow recommends bulk-lysis prior to staining for MRD analysis, while the authors use 50 ul of whole BM, so also there the statement "according to recommendations from the EuroFlow consortium" should be deleted.
- The data of the QA program look well, thank you for including this.
Author Response
- The EuroFlow AML/MDS panel is not designed for MRD. If the authors use it for MRD analysis, they must show data on the sensitivity. How was the sensitivity defined? How often was 0,1% reached, or 0,01%? Also it should be indicated how often tube 1 to 4 was used or if in some patients only a limited number of tubes was used (and if so, why).
Based on the reviewer's recommendation we have included the following information in the methods section:
The phrase “According to recommendations from the Euroflow consortium” has been deleted
MRD included in this study achieved a sensitivity of 0.1% and more than 90% was 0.01%. The level of sensitivity was considered based on the following conditions: number of acquired events, viable cells, patient's LAP if available, and bone marrow status (representation of bone marrow cells: mast cells, plasma cells, B precursors...).
Regarding the tubes used to MRD, the T1-T3 were used in more than 80% of patients. T4 of AML panel have been added when lymphoid markers are expressed, or the diagnostic phenotype was not available. We have added the following information:
For the design of the MRD, the different laboratories used panels from the myelodysplastic syndrome (MDS) panel of Euroflow (T1, T2, and T3 in more than 80% +/- T4 when lymphoid markers were expressed or the phenotype at diagnostic was not available) along with additional tubes depending on the patient's specific “leukemia-associated phenotype” (LAP). The tubes that include the LAP had to include the 4 backbones -HLADR, CD45, CD34 and CD117- together with another 4 antibodies at the discretion of each laboratory in accordance with those recommended by ELN for MRD6. When the phenotype of the blasts at diagnosis corresponded to monocytes, more than 1 tube associated with LAP was designed.
- Next to the EuroFlow tubes they use a LAIP tube. The authors should indicate which other markers (next to the backbone) are generally used (e.g. in a Supplemental Table).
In accordance with the reviewer's comment, we have added the following information in the methods section:
“The tubes that include the LAP had to include the 4 backbones -HLADR, CD45, CD34 and CD117- together with another 4 antibodies at the discretion of each laboratory in accordance with those recommended by ELN for MRD6.”
- The authors use a DfN and LAIP approach. It would be relevant and informative to show how these two approaches compare to each other. If the two approaches give discordant results, which data was used (e.g. highest of the two)?
For the study of MRD, panels of 3-4 tubes of 8 colors have been used that allow analyzing myeloid maturation. In addition, if the LAP was known, a designed tube was added for follow-up. However, to consider a positive MRD, it should be reproduced in the different tubes, and although it could have modified the diagnostic phenotype, if it presents a different phenotype than normal, it was considered positive MRD.
- If the authors focus on the multicenter aspect, they must provide data showing that the data from the 4 centers are comparable. They could for example show that patients characteristics are similar between the four centers and that also the distribution of MRD levels was similar between the four centers. Obviously it would be best if they could show that similar results were obtained in QA rounds, either based on fresh samples or electronic data files. Since centers used standardized protocols re-analysis of electronic data files should be relatively easy.
Considering the reviewer's suggestion, we have shared 6 MRD files to validate our reproducibility. We have added this information in supplementary files.
A comparative study of 6 MRD samples was carried out between the 4 centers. (Supplementary table 2).
Supplementary table 2. Comparative analysis of MRD samples in the 4 laboratories: The table represents the MRD value determined by each laboratory.
|
|
Lab 1 |
Lab 2 |
Lab 3 |
Lab 4 |
|
% MRD 1 |
0,54 |
0,55 |
0,39 |
0,54 |
|
% MRD 2 |
0,024 |
0,03 |
0,04 |
0,022 |
|
% MRD 3 |
0,27 |
0,24 |
0,23 |
0,18 |
|
% MRD 4 |
0,028 |
0,02 |
0,03 |
0,023 |
|
% MRD 5 |
0,002 |
0,003 |
0,005 |
0,002 |
|
% MRD 6 |
0,004 |
0,005 |
0,01 |
0,005 |
MRD: measurable residual disease
- Related to the above, it is still not clear how reproducible the data analysis is, adding such data would be very helpful.
- Since the EuroFlow AML/MDS panel is not designed for MRD, the authors should not refer to it as "according to recommendations from the EuroFlow consortium". They may indicate that instrument settings were performed according to EuroFlow and the the EuroFlow AML/MDS panel was used. It should also be noted that EuroFlow recommends bulk-lysis prior to staining for MRD analysis, while the authors use 50 ul of whole BM, so also there the statement "according to recommendations from the EuroFlow consortium" should be deleted.
According with the recommendation we have change that information.
“According to recommendations from the Euroflow consortium, Fifty µL of BM per tube were stained with the monoclonal antibodies. After 30 min of incubation at room temperature in the dark, erythrocytes ware lysed, and sample was washed.”
The current study has been conducted in the routine clinical practice, in real life, outside a specific trial in 4 centers that had to meet certain requirements. The sampling processing and the analysis was conducted according to the Euroflow protocols and recommendations, and Three laboratories did participate in the quality control organized within the Euroflow QA program.
- The data of the QA program look well, thank you for including this.
Reviewer 4 Report (Previous Reviewer 3)
The authors have adequately addressed my concerns
Author Response
The authors have adequately addressed my concerns
Thank you for your review
Round 2
Reviewer 2 Report (New Reviewer)
No concerns anymore.
Author Response
Thanks for your review
Reviewer 3 Report (Previous Reviewer 1)
Thank you for your reply and explanations. Although the paper has been improved, there still are some issues that need more detailed information:
- Authors claim a sensitivity of 0,01% in 90% of patients. They must provide data that show that they indeed reach this level of sensitivity. The sensitivity is not just a matter of BM quality and number of acquired events, but also depends a.o. on aberrant immunophenotype of the leukemic cells, the background in normal BM or BM during therapy, and the experience of the analyst. Were the labelings checked for their background in normal BM samples? Did the authors perform dilution experiments? How many cells were minimally required to consider something as positive?
- It will be informative if the authors more specifically indicate which markers were used in the LAIP approach rather than just referring to ELN. For example in a supplementary table showing the number of cases evaluated with a specific marker (or marker combination). In this way the readers can see what type of aberrancies were present and monitored.
- The authors use a DfN and LAIP approach but it remains unclear to me how they analyzed the data. It would be informative if the authors provide details about the concordance between the DfN approach (ideally split apart for the 4 tubes) and the LAIP tube, e.g. a table with concordance between the various tubes. If I understand it correctly, presence of MRD has to be confirmed in multiple tubes before it is considered as MRD positive. For the suggested table, the MRD results of the individual tubes should be shown.
Author Response
Thank you for your reply and explanations. Although the paper has been improved, there still are some issues that need more detailed information:
- Authors claim a sensitivity of 0,01% in 90% of patients. They must provide data that show that they indeed reach this level of sensitivity. The sensitivity is not just a matter of BM quality and number of acquired events, but also depends a.o. on aberrant immunophenotype of the leukemic cells, the background in normal BM or BM during therapy, and the experience of the analyst. Were the labelings checked for their background in normal BM samples? Did the authors perform dilution experiments? How many cells were minimally required to consider something as positive?
As the reviewer rightly points out, sensitivity does not only depend on the events acquired. We considered the marrow evaluable for analysis if mast cells, red series and less than 80% mature neutrophils were found. A cluster of 20 cells with phenotypic abnormalities was required for the detection of MRD. Hypocellular samples after chemotherapy administration are not considered valid for follow-up.. We did not perform dilution analysis. The Euroflow panel diagnostic tubes used have been previously evaluated in normal samples and in regeneration in each lab. Regarding the experience of the analyst, it is worth noting that two of the authors of this article are the reference analysts for the NOPHO AML protocol for children and 3 of the 7 reference laboratories for the PETHEMA AML protocols.
We have added the following information in methods section:
“We considered the marrow evaluable for analysis if mast cells, red series and less than 80% mature neutrophils were found. A cluster of 20 cells with phenotypic abnormalities was needed for detection of MRD.”
- It will be informative if the authors more specifically indicate which markers were used in the LAIP approach rather than just referring to ELN. For example in a supplementary table showing the number of cases evaluated with a specific marker (or marker combination). In this way the readers can see what type of aberrancies were present and monitored.
Based on the reviewer's recommendation we have added a supplementary table 2 with the designed panels.
In 240 cases tubes were added to the maturation tubes of the Euroflow MDS diagnostic panel. The combinations for LAIP analysis are shown in supplementary table 2.
Supplementary table 2: Designed panels to LAIP analysis
|
Combinations |
Backbone |
Different markers |
|
1 |
HLADR, CD45, CD34, CD117 |
CD15, CD13 or CD123, CD33, CD38 |
|
3 |
CD15, NG2, CD33, CD14 |
|
|
4 |
CD64, CD56, CD33, CD14 |
|
|
5 |
CD7, CD13, CD33, CD38 |
|
|
6 |
CD15, CD56, CD123, CD4 |
|
|
7 |
nuTdT, CD56, CD123, CD4 |
|
|
8 |
CD64, CD56, CD300e, CD14 |
|
|
9 |
CD15, CD33 or CD13, CD19, CD38 |
|
|
10 |
CD15, CD33 or CD13, CD7, CD38 |
|
|
11 |
CD36, CD13, CD11b, CD71 |
|
|
12 |
CD15, CD56, CD33, CD38 |
|
|
13 |
nuTdT, CD13, CD33, CD38 |
|
|
14 |
CD7, CD56, CD33, CD38 |
|
|
15 |
CD16, CD13, CD33, CD10 |
|
|
16 |
CD36, CD64, CD33, CD71 |
|
|
17 |
CD15, CD13, CD22, CD38 |
- The authors use a DfN and LAIP approach, but it remains unclear to me how they analysed the data. It would be informative if the authors provide details about the concordance between the DfN approach (ideally split apart for the 4 tubes) and the LAIP tube, e.g. a table with concordance between the various tubes. If I understand it correctly, presence of MRD has to be confirmed in multiple tubes before it is considered as MRD positive. For the suggested table, the MRD results of the individual tubes should be shown.
As stated in ELN recommendations, LAIPs are DfN abnormalities in the vast majority of cases, and the difference between these 2 approaches is likely to disappear if an adapted, sufficiently large panel of antibodies is used. Following these recommendations, a core panel was used to quantify MRD burden, detect emerging abnormalities and monitor patients lacking a diagnostic study.
There is no indication to quantify MRD in several tubes, except in the case of non-overlapping abnormalities. We used the described approach to reach a conclusion, based on the full information available, and this final conclusion (MRD negative or positive) is the result of integrating the DfN and LAIP analysis.
Unfortunately, the suggestion of the reviewer about the inclusion of “one table with the concordance between the DfN approach (ideally split apart for the 4 tubes) and the LAIP tube (a table with concordance between the various tubes in which the MRD results of the individual tubes should be shown)” is not possible due the time frame required to answer that suggestion, since it would require the review of all tubes of each case included in the study.
However, we would like to emphasize that the approach used in this work, combining the information of DfN and LAIP analysis to reach a conclusion, has been widely used in different articles, in which such tables have not been included.
Round 3
Reviewer 3 Report (Previous Reviewer 1)
Thank you for your revision.
Remaining comments:
- Supplemental Table 2: please indicate how often the various tubes were used (in how many patients).
- I agree that the DfN and LAIP approach in the end do the same thing. Nevertheless, it would still be interesting to compare the contribution of each tube to the final results, and to see how often the standard EuroFlow tubes did the job and how often the extra tube was needed for detection of MRD. Such information will be very helpful for other MRD labs and will significantly add to the relevance of this publication.
Author Response
Remaining comments:
- Supplemental Table 2: please indicate how often the various tubes were used (in how many patients).
As pointed out in line 114, in all patients we used the tubes corresponding to EuroFlow recommendation and in 240 an additional tube was used to determine LAIP. As stated in one of the responses to the reviewer's questions in a previous review, we used the described approach to reach a conclusion, integrating the results of all the tubes used for each case. Since this is a retrospective analysis, unfortunately, the information suggested by the reviewer can’t be included, since this table would require the review of all tubes of each case included in the study.
In line with our previous answers, we would like to emphasize that this type of information is not included in different MRD articles already published in Cancers as well as in other prestigious journals in which, occasionally, tables with information about tubes used for MRD assays are available in supplementary information, but the specific information required by the reviewer (“how often the various tubes were used (in how many patients”) has not been required for publication..
- I agree that the DfN and LAIP approach in the end do the same thing. Nevertheless, it would still be interesting to compare the contribution of each tube to the final results, and to see how often the standard EuroFlow tubes did the job and how often the extra tube was needed for detection of MRD. Such information will be very helpful for other MRD labs and will significantly add to the relevance of this publication.
We agree on the interest of such a question, and it might be the aim for futures studies. Nevertheless, as above indicated, we would like to emphasize that this is a retrospective study and, in order to be able to answer this question, we would need to go back to each patient's file to obtain the data from all centers which would take months. For this study, the database just includes the cuantification of MRD, this is, we have the information required to properly face the aim of this study, which is to evaluate the prognostic value of the quantification of MRD by flow cytometry before and after transplantation, but this study was not designed to answer the question of “in how many cases the EuroFlow requires additional information” and, therefore, we do not have that data available in the database.
We hope this limitation does not preclude the publication of this study in its current form. Also, we would like the reviewer and the editor to consider this article, and the information already added following the reviewers indications, in the context of the available literature from Cancers and other prestigious journals, in which such information is not included. We hope that, in its present form, the article will be of help for other MRD labs and clinicians while the aim proposed by the reviewer should the faced in future studies prospectively designed to answer that question.
Round 4
Reviewer 3 Report (Previous Reviewer 1)
-
This manuscript is a resubmission of an earlier submission. The following is a list of the peer review reports and author responses from that submission.
Round 1
Reviewer 1 Report
Caballer-Velazquez and colleagues report a multicenter study on flowcytometric MRD detection in AML patients undergoing allogeneic transplantation. They show that MRD levels prior to SCT have prognostic significance. and that also high MRD levels at day +100 after SCT are associated with poor prognosis. The paper generally reads well .
The authors indicate in the abstract that "... multicenter and standardized studies are lacking" and present their data obtained in four centers, indicating that they work according to the recommendations from the EuroFlow consortium. My major comments are related to this topic:
1. The Methods state: MRD was analyzed using 8-color panels based on EuroFlow protocols. EuroFlow has no AML MRD panel, therefore it is unclear what panel or which tubes were used. It is also not clear or truly EuroFlow tubes were used or whether changes were made with respect to clones, fluorochromes, or antibody combinations. Did all 4 centers use a fully identical panel or were there (minor or major) differences? A better description of the antibody panel(s) used should be included and the details of the panel(s) should be provided in the supplement. Based on the data in Suppl Figure 1, it seems that the authors made significant changes in the EuroFlow AML panel (CD38 FITC instead of APC-H7, combinations of CD64-CD33, CD7-CD13, ...).
2. As indicated above, EuroFlow has no AML MRD panel. If the authors used EuroFlow tubes, they should explain how they analyzed the data. They also mention that each center has their own criteria; what were the differences between the centers?
3. All four centers participated in the EuroFlow QA program. It would be good to indicate how they scored in this QA (information per fluorochrome), and such information can be added to the supplement.
4. The authors must proof that the data obtained in the four centers are comparable. Did they exchange samples or data files for analysis by other labs? Did they have meetings to discuss the methods and analysis and to standardize these? Was the distribution of MRD levels comparable between the four centers?
5. Sensitivity: the authors report MRD levels down to 0,01%. How did they define the sensitivity? Did they test their panels on normal and regenerating bone marrow samples? How reproducible were these low MRD levels if analyze by a second lab/person?
Other comments include:
- The authors identify three groups; MRD-neg, MRD-low and MRD-pos. However, these 3 groups are not always used or shown. I would either consistently show the data from the 3 groups throughout the paper, or combine the MRD-neg and MRD-low group.
- There are 43 patients who were MRD-pos but did not relapse, many of them had GVHD. It would be informative if these numbers of GVHD are compared with the remaining patients, to support the possible contribution of GVHD in preventing relapse.
Minor comments:
- Table 1: it may be informative it it is mentioned that the MRD data refer to pre-SCT time point.
- In the paper, a "," or "." is used for separating decimals, please use one of these consistently.
- Text says :Thirteen patients had MRD <0.1 (page 9), but in the table (suppl Table 3) there are 14 such patients.
Reviewer 2 Report
This is an interesting and well writen paper reporting the value of MRD measured by multiparameter flow cytometry before hematopoietic stem cell transplantation and at day 100 post-transplant.
Major comments:
* There is here too many information leading to a paper hard for the reader. I would suggest to divide it into two manuscripts: one focussing on the value of MRD before transplantation and one focussing on the value of MRD at day 100 post-transplant. Data regarding MRD at day 100 are to our opinion the most valuable.
Minor comments:
* Define all abbreviations in the abstract section.
* The panel of used markers should be indicated in the Materials and Methods section. Does it varied with time, the retrospective study going from 2012 to 2020?
* The authors should explain why they choose day 100 post-transplant for their MRD control. After transplant, MRD is generally checked after one month and then each 3 months.
* Was there any treatment received between transplant and day 100 post-transplant according to results of chimerism or molecular biology? Did some protocols used systematic administration of DLI?
* It is not clear why 295 on 318 allografted patients were included.
* The 10 patients allografted while in active disease should be removed from the study.
* In Table 1, column Global: Why not taking only into account the 295 patients included in the study?
* When considering MRD, the authors should be clearer and indicate if it concerns MRD before transplant or MRD at day 100.
* It would be interesting to analyze MRD at day 100 separately in patients with MRD positive before transplantation and in patients with MRD negative before transplantation.
* How about comparison between results of MRD measured by flow cytometry and MRD measured by molecular biology in patients with specific markers. Similar question between MRD measured by flow cytometry and measure of chimerism.
* Limitations of the study should be indicated in the discussion section: Retrospective study, time of study from 2012 to 2020, heterogeneity of conditioning treatment, heterogeneity of leukemic stage.....
* In Supplementary Table 2, the authors have forgotten spanish words.
* Was there any relationship between MRD measured at day 100 and the intensity of GvHD?
* References should be checked. They are not all presented on the same way.
Reviewer 3 Report
In this article, the authors (Caballero-Velázquez et al) aimed to evaluate through a retrospective analysis the prognostic impact of MRD assessed by multiparametric flow cytometry (MFC) in patients with AML undergoing HSC transplantation, with a major focus on OS and LFS.
It is now well established that MRD assessment is a key parameter in the prognostic stratification of patients with AML, and several studies have already explored its role before and after HSCT, many using MFC.
The present study, nonetheless, is interesting because MFC was conducted accurately using equipment and analyses standardized according to Euroflow protocols and ELN recommendations.
Some points of concern, however, should be addressed.
Major Points:
1) Only 295 of the 318 patients considered were included in the analysis. It is not clear why 23 pts were excluded (lack of MRD data?). This point should be carefully explained, and the characteristics of these patients described. Besides, in table 1 and 2 the global population of 318 patients is considered. I suggest, since only 295 were analyzed, to consider only these pts in the comparison. Otherwise, better clarify why you made this choice.
2) Surprisingly, age did not emerge as a significant independent factor for OS and NRM. Could you elaborate more on this point?
3) I would elaborate more on MRD dynamics. Could you describe the outcome of MRDpos/MRD- pre/post HSCT compared to the remining ones? As you correctly discussed, this aspect can be of interest.
4) It would have been interested to compare MFC data with molecular ones (either qPCR and/or NGS), as other groups have done in the past. Are these data available for some patients at least (eg NPM1?). If so, please perform this analysis. Otherwise, clearly acknowledge this relevant limitation.
5) Data on induction/consolidation therapy could be of interest, as innovative drugs and/or mor intensive regimens might play a role. Did all patients receive the same therapy according to risk class? Did some receive specific targhet therapies, e.g., FLT3 or IDH2 inhibitors?
Minor points
1) One of the pivotal studies on the subject is not cited “Walter RB, Impact of pretransplantation minimal residual disease, as detected by multiparametric flow cytometry, on outcome of myeloablative hematopoietic cell transplantation for acute myeloid leukemia. J Clin Oncol. 2011 Mar 20;29(9):1190-7. “. Please include it
2) You correctly consider ELN risk stratification in the univariate and multivariate analysis. What classification did you consider? I guess 2017, please clarify
3) To classify pts according to ELN 2017 classifications, NGS data are required in those lacking other stratifying lesions. Can you confirm those data are available? Please clarify the number of missing.
4) The numbers in table 1 are confusing, as pts with missing data are not detailed. Besides, please check “disease status at transplantation” (272+29+28+15+6=350?)
5) Avoid commas to separate decimals (eg line 36, Table 3,line 219, fig 2 …..)
6) In the tables, please omit the “ %” while describing the percentages. It is not useful, since it is already explicated in the left column [ n (%) ]. Besides, a comma should be added in the left column [eg Acute GVHD, n (%) ]
7) Please rephrase lines 171-173 to avoid repetitions
8) Line 174: I would prefer “that is” or “namely” rather than “this is”
9) Please omit the supplementary tables form the main text, and improve their layout
10) In supl Table 2 HR is 1 and CI include 0 [95% CI 0.98-1.03]but p appears 0.001. Please clarify
11) Line 296: “sight” ?
12) Line 297: the “s” of patients is missing